# Identification of Potential Sex-Specific Biomarkers in Pigs with Low and High Intramuscular Fat Content Using Integrated Bioinformatics and Machine Learning

**DOI:** 10.3390/genes14091695

**Published:** 2023-08-25

**Authors:** Yongli Yang, Xiaoyi Wang, Shuyan Wang, Qiang Chen, Mingli Li, Shaoxiong Lu

**Affiliations:** Faculty of Animal Science and Technology, Yunnan Agricultural University, Kunming 650201, China; 15987169785@163.com (Y.Y.); wangxiaoyi0101@126.com (X.W.); shuyanwang2014@126.com (S.W.); chq@sjtu.edu.cn (Q.C.); xiaolucao@126.com (M.L.)

**Keywords:** pig, intramuscular fat content, integrated bioinformatics, machine learning, sex-specific biomarker

## Abstract

Intramuscular fat (IMF) content is a key determinant of pork quality. Controlling the genetic and physiological factors of IMF and the expression patterns of various genes is important for regulating the IMF content and improving meat quality in pig breeding. Growing evidence has suggested the role of genetic factors and breeds in IMF deposition; however, research on the sex factors of IMF deposition is still lacking. The present study aimed to identify potential sex-specific biomarkers strongly associated with IMF deposition in low- and high-IMF pig populations. The GSE144780 expression dataset of IMF deposition-related genes were obtained from the Gene Expression Omnibus. Initially, differentially expressed genes (DEGs) were detected in male and female low-IMF (162 DEGs, including 64 up- and 98 down-regulated genes) and high-IMF pigs (202 DEGs, including 147 up- and 55 down-regulated genes). Moreover, hub genes were screened via PPI network construction. Furthermore, hub genes were screened for potential sex-specific biomarkers using the least absolute shrinkage and selection operator machine learning algorithm, and sex-specific biomarkers in low-IMF (troponin I (*TNNI1*), myosin light chain 9(*MYL9*), and serpin family C member 1(*SERPINC1*)) and high-IMF pigs (*CD4* molecule (*CD4*), *CD2* molecule (*CD2*), and amine oxidase copper-containing 2(*AOC2*)) were identified, and then verified by quantitative real-time PCR (qRT-PCR) in semimembranosus muscles. Additionally, the gene set enrichment analysis and single-sample gene set enrichment analysis of hallmark gene sets were collectively performed on the identified biomarkers. Finally, the transcription factor-biomarker and lncRNA-miRNA-mRNA (biomarker) networks were predicted. The identified potential sex-specific biomarkers may provide new insights into the molecular mechanisms of IMF deposition and the beneficial foundation for improving meat quality in pig breeding.

## 1. Introduction

Intramuscular fat (IMF) content is an important trait in the pig industry, which is positively correlated with meat quality, and affects the tenderness, flavor, and juiciness of the meat [1,2]. The potency of fatty acid metabolism in skeletal muscle and the development of intramuscular adipocytes have a significant impact on IMF content [3]. Many studies have shown that the pattern of muscle development and IMF deposition depended on breed, genotype, sex, muscle location, diet, and slaughter weight characteristics [4,5,6,7], but the related molecular mechanisms remain elusive.

In recent years, great progress has been made in genetic mechanisms and candidate gene exploring for IMF content based on advanced molecular biology techniques and bioinformatics. Many genes have been identified as major regulators of IMF deposition, playing important roles in regulating fat metabolism and promoting IMF deposition [8]. For example, acetyl-CoA Acyltransferase 1(*ACAA1*) is involved in fatty acid oxidation and lipid metabolism, and is strongly linked to *PPAR* signaling and fatty acid metabolism pathways [9]. Acyl-CoA Synthetase Long Chain Family Member 4(*ACSL4*) positively regulates adipogenesis in IMF cells, and its high expression can increase monounsaturated and polyunsaturated fatty acid contents in pig intramuscular adipocytes [10]. The higher fat deposition in Alentejano pigs (obese type) than in Bisaro Portuguese pigs (lean type) may be due to the increased synthesis of new fatty acids caused by the up-regulation of key genes such as ATP Citrate Lyase (ACLY), fatty Acid Synthase (FASN) and malic Enzyme 1(ME1) [11]. However, most studies have used pigs of different breeds or different IMF contents within the same breed [12,13], with few studies on individuals of different sexes in the same breed to identify candidate genes and molecular mechanisms. Regarding sex traits, previous studies have shown significant differences in nearly all carcass characteristics between castrates and gilts; castrates produce fattier carcasses than gilts, which deposit leaner meat, and IMF levels vary considerably between sexes at the physiological and biochemical levels [14]. The results of studies by Latorre et al. and Serrano et al. showed that the IMF content and marbling of castrates were considerably higher than those of gilts, producing, to some extent, better meat quality traits [15,16]. Previous studies have also found significantly higher total fatty acid levels in castrates than in gilts for the fatty acid composition of IMF and semimembranosus muscles [17]. Numerous studies have demonstrated a significant influence of sex on pork quality, with discernible differences in IMF content between male and female pigs [18,19]. However, most studies for the differences in pork quality and IMF content between males and females did not involve a molecular and regulatory mechanism.

Consequently, our study aimed to screen potential sex-specific biomarkers in male and female pigs with different IMF levels in the same breed using integrated bioinformatic methods and machine learning algorithms. In addition, we aimed to construct molecular regulatory networks related to sex-specific biomarkers, laying a foundation for in-depth exploration of the molecular mechanisms underlying IMF deposition.

## 2. Materials and Methods

The research flowchart of the present analysis is shown in Figure 1.

### 2.1. Data Collection and Preprocessing

The GSE144780 gene expression dataset was retrieved from the public Gene Expression Omnibus database [GPL19176 Illumina HiSeq 2000 (*Sus scrofa*) (https://www.ncbi.nlm.nih.gov/geo/), accessed on 1 November 2022] and generated using semimembranosus muscles (SM) from Italian Large White pigs (six males and six females) [12]. The pigs were selected from 950 purebred sib-tested Italian Large White pigs, reared in the same environmental conditions on the same finishing diet and fed quasi *ad libitum*. Based on the extreme and divergent contents of SM IMF, twelve pigs were categorized into two groups: 6 low- and 6 high-IMF animals. In our study, the individuals with different IMF levels in the two groups were evenly distributed in the two sex subgroups, each with 3 male and 3 female pigs in the low- and high-IMF group. Details about the individuals are shown in Table 1. The original study has been approved by the Institutional Review Board of the relevant participating institutions (European rules (Council Regulation (EC) No. 1/2005 and Council Regulation (EC) No. 1099/2009)), and no additional approval was required from the ethics committee in dataset GSE144780.

### 2.2. Identification of Differentially Expressed Genes (DEGs)

DEGs between males and females in the low- and high-IMF groups were identified separately using the “Deseq2” package of the R software (version 4.1.2) [20]. In this study, the genes with a |log2FC (fold change)| > 1 and a *p*-value < 0.05 were considered as DEGs. The expression level and distribution of DEGs were visualized using the heatmap and the volcano map based on the ggplot2 package (versions 3.3.6), respectively.

### 2.3. DEG Functional Enrichment Analysis

Functional DEG analysis was implemented via a GO and KEGG pathway analysis using the online DAVID database (DAVID 2021, https://david.ncifcrf.gov/home.jsp, accessed on 1 November 2022). The GO term included three categories: the cellular component (CC), biological process (BP), and molecular function (MF). The functional terms with a *p*-value < 0.05 were considered to significantly change, and the functional terms were visualized using the R-package ggplot2.

### 2.4. Protein–Protein Interaction (PPI) Network Construction and Hub Gene Identification

Functional interactions between DEG-encoded proteins were analyzed by constructing a PPI network using the STRING online database (https://cn.string-db.org/, accessed on 1 November 2022) with a combined score > 0.4 and *p*-value < 0.05, shown by the Cytoscape software (version 3.9.1) [21]; each node represented the gene-encoding protein and edges represented the connection between nodes in the PPI network. Subsequently, the cluster analysis was performed in the whole PPI network using the Molecular Complex Detection (MCODE) algorithm by the Cytoscape plugin (Version 2.0.2) [22]. The threshold parameters were set for the degree cutoff = 2, node score cutoff = 0.2, k-score = 2, and max. depth = 100. The hub genes were identified using the Maximal Clique Centrality (MCC) method in the Cytoscape plugin cytoHubba (Version 0.1) [23], and the top 10 genes with the higher MCC scores were selected as hub genes. The interaction network of the hub genes and GO and KEGG terms were visualized by ClueGo (Version 2.5.9) and CluePedia plugin (Version 1.5.9). Thereafter, the correlations of hub genes were analyzed and visualized by the Pearson method through the ggpairs of GGally (version: 1.5.0) package based on R software.

### 2.5. Screening of Potential Sex-Specific Biomarkers

The least absolute shrinkage and selection operator (LASSO) machine learning algorithm [24] was used to screen for potential sex-specific biomarkers among the top 10 hub genes identified in the low- and high-IMF groups. In addition, LASSO was built with the “glmnet” R package and performed variable screening and complexity adjustment while fitting a generalized linear model. To evaluate the predictive performance of potential sex-specific biomarkers, their expression levels were compared between the males and females using an unpaired *t*-test. *p* < 0.05 was considered to be significant. Then, the receiver operating characteristic (ROC) curves of dataset GSE144780 (source set) were established, and the area under the curve (AUC) values of potential biomarkers was calculated to assess the efficacy of the distinguishing performance of potential sex-specific biomarkers using GraphPad Prism (version 8.0.1).

### 2.6. Analysis of Hallmark Gene Sets of Potential Sex-Specific Biomarkers

To further elucidate the functions of the potential sex-specific biomarkers, an association analysis of the hallmark gene sets was performed using a single-sample gene set enrichment analysis (ssGSEA) [25]. Firstly, we calculated the ssGSEA scores of hallmark gene sets as well as in the males and females, and subsequently analyzed the association between hallmark gene sets and potential sex-specific biomarkers; the “corrplot” package was used to obtain the Spearman rank correlation coefficient.

### 2.7. Construction of Potential Transcription Factor (TF) Sex-Specific Biomarker Regulatory Network

TFs of potential sex-specific biomarkers were predicted using the online database of UCSC and JASPAR 2022 (https://genome.ucsc.edu/, accessed on 1 November 2022), and the minimum score = 600 was the screening condition. The regulatory relationships between TFs and potential sex-specific biomarkers were visualized through the Cytoscape software (version 3.9.1).

### 2.8. Construction of ceRNA Network

The lncRNA-miRNA-mRNA interactive relationships were elucidated by constructing a lncRNA-miRNA-mRNA (potential sex-specific biomarkers) regulatory network based on the ceRNA hypothesis [26]. The potential targeted miRNAs and lncRNAs of potential sex-specific biomarkers were predicted by the online ENCORI database (https://rna.sysu.edu.cn/encori/index.php, accessed on 1 November 2022). First, the biomarkers-miRNA relationship pairs were extracted based on at least two databases: miRanda [27], TargetScan [28], RNA22 [29], and miRmap [30]. Then, the interacting lncRNAs were predicted based on above-predicted miRNAs. Finally, the co-expression network of the lncRNAs-miRNAs-mRNA (potential sex-specific biomarkers) ceRNA network was visualized through the Cytoscape software.

### 2.9. Animals and Tissue Collection

Thirty pigs, including fifteen males and fifteen females of the Saba pig, were used in this study, which is an indigenous pig breed in Yunnan, China. The pigs used in this study were obtained from the national-level Saba pig conservation farm (Chuxiong City, Yunnan, China). The pigs were selected from purebred Saba pigs, raised in the same environmental conditions with free access to water and feed until slaughter (~100 kg). The SM samples were collected from the carcasses at slaughter, snap frozen in liquid nitrogen, and maintained at −80 °C until total RNA extraction; some SM was stored at −20 °C for the determination of IMF content. All animal procedures were performed according to the principles of the Laboratory Animal Care and Use Guidelines issued by the Animal Research Committee of Yunnan Agricultural University; the experimental protocol was approved by the Animal Ethics Committee of Yunnan Agricultural University (approval ID: YAUACUC01).

### 2.10. Measurement of IMF Content

The IMF content of SM was measured by the Soxhlet extraction method [31]. Firstly, the SM sample was dried and crushed (weight W_1_ g), and then transferred to the extraction chamber of the soxhlet apparatus and soaked in anhydrous ether overnight. Subsequently, the anhydrous ether backflow equipment was worked for 8 h at 80 °C. Finally, the remaining sample was dried again and weighed (weight W_2_ g). The IMF content of SM was calculated using the following equation: IMF (%) = [(W_1_ − W_2_)/W_1_] × 100. Considering the weight and IMF content as factors, three male pigs and three female pigs exhibiting exceptionally high IMF content and three male pigs and three female pigs displaying remarkably low IMF content were selected for the subsequent validation analysis.

### 2.11. RNA Extraction and qRT-PCR

The total RNA was extracted from SM samples using an RNA sample total Extraction Kit (Tiangen, Beijing, China). Briefly, the SM sample was frozen in liquid nitrogen and ground to a fine powder. Then, chloroform was added, the aqueous phase was collected and translated to RNase-Free columns, and the RNA was precipitated by the addition of absolute ethanol. Subsequently, buffer RD and RW were stepwise added to an RNA pellet to obtain RNA. Finally, RNA was eluted from the column by 50 µL of RNase-free water. Reverse transcription was performed using PrimeScript™ RT reagent Kit with gDNA Eraser (Takara, Dalian, China) according to the manufacturer’s instructions. The first step considers: the reaction conditions of system1: 42 °C, 2 min; 4 °C, 18 min; second, the reaction conditions: 37 °C, 15 min; 85 °C, 5 s; 4 °C, 18 min. qPCR assay was performed using TB Green^®^ Premix Ex Taq™ II (Tli RNaseH Plus) (Takara, Dalian, China) on a qPCR system (Mx3000P, Agilent Technologies, Santa Clara, CA, USA). The reaction system was prepared for qPCR reaction. It was performed using a two-step amplification, Stage1 (predenaturation): 95 °C, 30 s; Stage2: repeat 40, 95 °C, 30 s, 60 °C, 30 s; Stage3: Dissociation. The gene-specific qPCR primers are listed in Appendix A. Each experiment was performed in triplicates, and the relative expression of mRNA were calculated through the 2^−ΔΔCt^ method; GAPDH was used as the internal control for normalization. The relative mRNA expression levels of male and female pigs were compared by unpaired *t*-test using SAS software (version 9.2), *p* < 0.05 was considered significant, and *p* < 0.01 was considered extremely significant.

## 3. Results

### 3.1. DEG Identification

Totals of 162 DEGs were identified between male and female pigs in the low-IMF group (*p* < 0.05), 64 of which were significantly up-regulated, and 98 were significantly down-regulated in males (Figure 2A; Appendix A). A total of 202 DEGs were identified in the high-IMF group, 147 of which were significantly up-regulated, and 55 were significantly down-regulated in males (Figure 2C; Appendix A). The expression levels of these DEGs in the low- and high-IMF groups were presented using a heatmap in Figure 2B,D, respectively.

The up- and down-regulated DEGs between the low- and high-IMF groups were further compared, as shown in Figure 3E. The identified overlapping, up-regulated, and down-regulated DEGs in the low- and high-IMF groups were mentioned in Table 2. Among the overlapping DEGs, GADD45G-interacting protein 1 (*GADD45GIP1*) and inter-α-trypsin inhibitor heavy chain 1 (*ITIH1*) were up-regulated in high-IMF male pigs, but down-regulated in male pigs with low IMF.

### 3.2. DEG Functional Analysis

In the low-IMF group, GO analysis of DEGs of male and female pigs showed that 16 BPs, 5 CCs, and 7 MFs were significantly enriched (*p* < 0.05) (Appendix A). The top five GO terms with the smallest *p* values were shown in Figure 2F. The KEGG enrichment analysis of DEGs showed 12 significantly enriched pathways (*p* < 0.05) (Appendix A). The top 10 KEGG terms with the smallest *p* values were shown in Figure 2G. The up-regulated enriched terms were mainly related to muscle processes, such as ventricular cardiac muscle tissue morphogenesis, muscle contraction, transition between fast and slow fiber, and skeletal muscle contraction (Appendix A).

In the high-IMF group, GO analysis showed that a total of 5 BPs, 5 CCs, and 2 MFs were significantly enriched (Figure 2H, Appendix A). KEGG enrichment analysis of DEGs showed 15 significantly enriched pathways (Figure 2I, Appendix A). Up-regulated enriched terms were mainly related to immune-related processes, such as T cell receptor signaling pathway, JAK-STAT signaling pathway, and PI3K-Akt signaling pathway (Appendix A).

Only one of all enriched GO terms (GO:0010951: negative regulation of endopeptidase activity) was found to overlap in a functional enrichment analysis. The results also suggested that the molecular regulatory mechanisms of male and female pigs differ substantially between the low- and high-IMF groups.

### 3.3. PPI Network Construction

From the low-IMF group, 162 DEGs were submitted to the STRING database, which identified a PPI network with 152 nodes and 165 edges (Figure 3A). A highly correlated module analysis showed four modules in the whole PPI network with the highest score (score = 6) including 9 nodes and 24 edges (Figure 3B–E). Enrichment analysis findings revealed that these clusters’ genes were primarily enriched by signaling pathways including a fibrinolysis, glyoxylate and dicarboxylate metabolism; glycine, serine and threonine metabolism; and glycine catabolic process (Figure 3F–H).

From the high-IMF group, 202 DEGs were submitted to the STRING database, which identified a PPI network with 184 nodes and 150 edges (Figure 3L). One module (score = 7), including 9 nodes and 24 edges, was identified in the whole PPI network (Figure 3M). A cluster enrichment analysis revealed that functions were mainly associated with immune-related response and cell molecule adhesion, such as the positive regulation of interferon-γ/interleukin-2 production, positive regulation of T cell proliferation, regulation of cytokine biosynthetic process, and positive regulation of α-β T cell activation (Figure 3N).

### 3.4. Identification and Analysis of Hub Genes

In the low-IMF group, the top 10 genes with the highest MCC scores were *TNNI1* (score = 1092), myosin Light Chain 2 (*MLC2V*) (score = 1032), myosin heavy chain 6(*MYH6*) (score = 1008), myosin light chain 3(*MYL3*) (score = 888), troponin C1(*TNNC1*) (score = 774), myosin binding protein C3(*MYBPC3*) (score = 744), ryanodine receptor 2(*RYR2*) (score = 722), *MYL9* (score = 318), tropomyosin 3(*TPM3*) (score = 264), and *SERPINC1* (score = 230) in the whole PPI network (Figure 3I), which were identified as hub genes. Apart from *MYBPC3*, *RYR2*, *MYL9*, and *SERPINC1*, the six remaining hub genes were up-regulated in males, compared with females (Figure 2A). The expression correlations among the 10 hub genes were shown in Figure 3J. The interactive relationships between hub genes and GO and KEGG terms were shown in Figure 3K.

In the high-IMF group, the top 10 genes were *CD4* (score = 308), protein tyrosine phosphatase receptor type C (*PTPRC*) (score = 305), selectin l (*SELL*) (score = 289), *CD69* molecule(*CD69*) (score = 288), *CD3E* molecule(*CD3E*) (score = 240), *CD2* (score = 121), *AOC2* (score = 120), C-X-C motif chemokine receptor 4(*CXCR4*) (score = 38), serpin family B member 2(*SERPINB2*) (score = 26), and *MYC* proto-oncogene (*MYC*) (score = 15) in the whole PPI network (Figure 3O). They were identified as hub genes, which were all up-regulated in males (Figure 2C). The expression correlations among the 10 hub genes were shown in Figure 3P. The interactive relationships between hub genes and enrichment terms were shown in Figure 3Q.

### 3.5. Identification of Potential Sex-Specific Biomarkers

In the low-IMF group, the LASSO algorithm analysis revealed three candidate genes, *TNNI1*, *MYL9*, and *SERPINC1*, as potential sex-specific biomarkers (Figure 4A). The *TNNI1* expression was significantly up-regulated in males, compared with females, while *MYL9* and *SERPINC1* showed no significant differences in expression levels (Figure 4B–D). The ROC analysis based on the GSE144780 dataset showed that the AUCs of *TNNI1*, *MYL9*, and *SERPINC1* related to sex were 1, 1, and 0.8889 (AUC > 0.7), respectively, all of which showed a higher effectiveness. In particular, *TNNI1* may be the sex-specific biomarker with the highest potential in low-IMF pigs (Figure 4B–D). GSEA of the three potential biomarkers detected multiple pathways, including complement and coagulation cascades, cardiac muscle contraction, and a calcium signaling pathway consistent with the DEG functional analysis results. In addition, lipid metabolism pathways and myocardial tissue damage-associated pathways, such as the cholesterol metabolism, p53 signaling pathway, and butanoate metabolism pathway, were identified, indicating that lipid deposition and myocardial tissue statuses differ between males and females in low-IMF pigs. The GSEA results are shown in Figure 5A–C, Appendix A.

In the high-IMF group, three candidate genes, *CD4*, *CD2*, and *AOC2*, were identified as potential sex-specific biomarkers (Figure 4E). The *CD4* and *AOC2* expression was significantly up-regulated in males (Figure 4F–H). The AUCs of *CD4*, *CD2*, and *AOC2* related to sex were all =1, all showing a higher effectiveness. In particular, *AOC2* might be the sex-specific biomarker with the highest potential in high-IMF pigs (Figure 4F–H). GSEA analysis showed that potential sex-specific biomarkers in the high-IMF group were enriched in immune and inflammatory response-related pathways, such as the IL-17, and Toll-like receptor signaling pathways. Notably, males are more active than females in those pathways. In addition, lipid metabolism-related pathways, such as the regulation of lipolysis in adipocytes; *PPAR* signaling pathway and fat digestion and absorption; and immune/inflammation-mediated diseases, such as autoimmune thyroid disease and inflammatory bowel disease; were identified (Figure 6A–C, Appendix A). To summarize, we speculate that these sex-specific biomarkers may affect the IMF deposition by immune and inflammatory responses, as well as lipid metabolism.

### 3.6. Analysis of Hallmark Gene Sets in Difference Sex

In the low-IMF group, HALLMARK coagulation in males was lower than that in females, and HALLMARK_glycolysis was higher (Figure 7A). To the sex-specific biomarkers (Figure 7B), *TNNI1* was significantly associated with estrogen_response_late (*p* < 0.05); *MYL9* was positively correlated with complement, apoptosis, and androgen response (*p* < 0.05); and *SERPINC1* was not observed with significant correlation with hallmark gene sets (*p* > 0.05).

In the high-IMF group, there were no significant differences between males and females in hallmark gene sets (Figure 7C). As shown in Figure 7D, *AOC2* was negatively correlated with wnt_β_catenin_signaling (*p* < 0.05) and notch_signaling (*p* < 0.01) and positively associated with myc_targets_v2 (*p* < 0.05), myc_targets_v1 (*p* < 0.05), and mtotic1_signaling (*p* < 0.001). *CD2* was negatively correlated with wnt_β_catenin_signaling (*p* < 0.05) and positively associated with oxygen species pathway (*p* < 0.05), myc_targets_v1 (*p* < 0.05), and adipogenesis (*p* < 0.05). However, no significant correlations were observed between *CD4* and hallmark gene sets (*p* > 0.05)

### 3.7. Gene Regulatory Network Analysis of Potential Sex-Specific Biomarkers

In the low-IMF group, a total of 36 TFs were predicted based on potential sex-specific biomarkers, including six TFs predicted by *TNNI1*, 10 TFs predicted by *MYL9*, and 20 TFs predicted by *SERPINC1*. Subsequently, a potential sex-specific transcriptional regulatory network was constructed (Figure 4I). In the high-IMF group, a total of 12 TFs were predicted, including five TFs predicted by *CD4*, three TFs predicted by *CD2*, and four TFs predicted by *AOC2*. Finally, a TF-potential sex-specific biomarker network was constructed (Figure 4K).

### 3.8. CeRNA Network Analysis of Potential Sex-Specific Biomarkers

In the low-IMF group, a total of eight mRNA-miRNA pairs, including three mRNAs and eight miRNAs, were predicted to be related to three potential sex-specific biomarkers. The interacting lncRNAs were predicted according to these miRNAs, and six interaction pairs (including eight miRNAs and 16 lncRNAs) were predicted. Finally, the ceRNA network was constructed using Cytoscape (Figure 4J). In the high-IMF group, a total of six mRNA-miRNA relationship pairs, including three mRNAs and six miRNAs, were predicted. Then, 46 miRNA-lncRNA interaction pairs (including six miRNAs and 27 lncRNAs) were predicted, and a ceRNA network was constructed (Figure 4L).

### 3.9. IMF Content of SM in Saba Pigs

This set of samples consisted of 12 individuals, chosen from the population of thirty tested Saba pigs. For their extreme and divergent contents of SM IMF, the animals were divided into two groups, low- and high-IMF groups, each with six pigs (three males vs. three females), as shown in Table 3.

### 3.10. Validation of the Biomarkers via qRT-PCR

The expression trends for the potential sex-specific biomarkers *TNNI1*, *MYL9,* and *SERPINC1* in low-IMF pigs and *CD4*, *CD2,* and *AOC2* in high-IMF pigs in the SM tissues were consistent with the results of the transcriptome analysis. In addition, significant differences in mRNA expression levels of *TNNI1* (*p* = 0.0266) and *MYL9* (*p* = 0.0286) between the females and males in the low-IMF group were observed (Figure 4M), and *CD2* (*p* = 0.0009) and *AOC2* (*p* < 0.0001) in the high-IMF group were observed (Figure 4N).

## 4. Discussion

Pork is an important source of meat for human consumption, and IMF content is a key driver of meat quality and a polygenic characteristic in pigs. Previous studies have analyzed IMF-related hub genes and their functions [32,33,34], but the molecular mechanisms underlying IMF content still remain to be completely explored. In light of this, our study represented the pioneering application of bioinformatics methodologies to comprehensively investigate the molecular mechanisms underlying IMF deposition in male and female pigs with varying IMF levels within the same breed. The primary goals of the current work included hub gene screening, machine learning algorithm integration for potential sex-specific biomarker screening, functional analysis, and regulatory mechanism prediction of the obtained potential sex-specific biomarkers.

In the present study, 162 DEGs, including 64 up- and 98 down-regulated genes, were detected between males and females in the low-IMF group; 202 DEGs, including 147 up- and 55 down-regulated genes, were screened in the high-IMF group. Eleven overlapping DEGs were identified in low- vs. high-IMF pigs. Among the overlapping genes, *GADD45GIP1* and *ITIH1* were up-regulated in high-IMF male pigs, but down-regulated in low-IMF male pigs; thus, they might play opposite regulatory roles in male and female pigs with different IMF contents. *GADD45GIP1*, also known as adipocyte-specific Crif1, plays a vital role in regulating adipocyte oxidative phosphorylation function [35]. *ITIH1* is a member of the inter-α-trypsin inhibitor family of proteins that has been implicated in multiple inflammatory diseases. The ITIH family (*ITIH1*, *ITIH2*, *ITIH3*, and *ITIH4*) is considered crucial for maintaining the uterine surface glycocalyx during placental attachment in pigs, according to a prior study [36]. Up to now, studies of *GADD45GIP1* and *ITIH1* on pork IMF contents of different sexes have not been reported. Our study may provide a novel insight into the role of *GADD45GIP1* and *ITIH1* in IMF deposition between the sexes, which requires further exploration and confirmation.

### 4.1. Low-IMF Group Analysis

Functional analysis indicated that the DEGs in the low-IMF group were primarily associated with BPs, such as ventricular cardiac muscle tissue morphogenesis, biomineral tissue development, cardiac muscle contraction, and muscle contraction. These GO terms correspond with previous studies identifying DEGs related to modulating adipogenesis in intramuscular adipocytes [37], the impact of muscle contraction on intramuscular adipose tissue [38], muscle fiber composition [39], and fat deposition in pigs [40]. KEGG enrichment analysis revealed that these genes were mainly involved in immunity and inflammation-relevant pathways, including complement and coagulation cascades and platelet activation. Complement systems and coagulation cascades might play a vital role in post-trauma and subsequent inflammatory reactions as innate immunity elements [41]. A previous study observed greater IMF deposition in Korean cattle after castration, which might lead to alterations in the complement and coagulation cascade pathways, possibly owing to obesity and the subsequent inflammation [42]. Platelet activation is regulated by the complement system, and platelet activity is mostly linked to the start of coagulation cascades [43].

Subsequently, 10 hub genes were selected from the PPI network via the topology analysis method—MCC, and were mainly involved in muscle fiber and tissue development-associated processes, especially in the development of muscle organs, such as cardiac chamber morphogenesis, ventricular cardiac muscle tissue development, and cardiac muscle tissue morphogenesis; this was consistent with the results of our previous functional analysis of DEGs. Furthermore, *TNNI1*, *MYL9*, and *SERPINC1* genes were identified as potential sex-specific biomarkers in low-IMF pigs using the LASSO algorithm. Among them, *TNNI1* might be the sex-specific biomarker in low-IMF pigs with the highest potential. The functional analysis of *TNNI1* also indicated that it might be involved in the muscle regulatory response. *TNNI1* is a slow-twitch skeletal muscle isoform expressed solely in slow-twitch skeletal fibers and is related to the muscle fiber type. A previous study has shown that *TNNI1* gene polymorphisms were related to IMF content and meat color in the biceps femoris of pigs [39]. Additionally, abundant *TNNI1* expression was detected in L-arginine-supplemented pigs [44], suggesting that *TNNI1* might play a positive regulatory role in IMF content and meat quality improvement. Abundant *TNNI1* expression was also found in cardiac muscles [45]. One study demonstrated that *TNNI1* could act as a marker of pork quality [46]. In our study, *TNNI1* was expressed at a higher level in males, and it had a significant positive correlation with *TPM3*, *TNNC1*, and *TPM3* expression levels, which were related to muscle formation and development [47,48,49]. *MYL9*, which encodes myosin light chain, may regulate muscle contraction by modulating the ATPase activity of myosin heads. Previous studies have shown that *MYL9* was correlated with ATP kinase activity regulation, muscle cell control, and signal transduction [50]. Several studies have found that *MYL9* displayed higher transcriptome and proteome expression levels in myostatin-knockout Meishan pigs, and it was hypothesized that *MYL9* might play an important role in skeletal muscle growth and development [51]. *SERPINC1*, a member of the serpin superfamily, contributes to the regulation of the blood coagulation cascade. A previous study showed higher *SERPINC1* expression in female vs. male livers due to the different growth hormone secretion patterns between the sexes [52]. A recent study showed that *SERPINC1* played a critical role in immune responses [53]. *SERPINC1* expression is inherently associated with immune cell infiltration by B cells, T cells, CD4^+^ T cells, macrophages, neutrophils, and dendritic cells [54]. Reports have shown that the *SERPINC1* gene was essentially related to fat metabolism and development during beef cattle growth [55]. Although no studies have reported a direct correlation between *TNNI1*, *MYL9*, *SERPINC1* and muscle formation, and IMF deposition in male and female low-IMF pigs, we speculated that these biomarkers might be involved in advancing IMF deposition through their involvement in muscle development based on previous studies. Our results also further indicate that *TNNI1*, *MYL9,* and *SERPINC1* are closely associated with muscle development and IMF content.

The GSEA functions of *TNNI1*, *MYL9*, and *SERPINC1* included muscle development-related and immunity-related pathways, such as cardiac muscle contraction, calcium signaling pathway, complement and coagulation cascades, and bile secretion, which are consistent with the DEG functional analysis results described above. We also identified energy metabolism-related signaling pathways, such as the cholesterol metabolism, cAMP signaling pathway, TCA cycle, and p53 signaling pathway. CAMP, synthesized from ATP by the adenylate cyclase family, played a crucial role in both adipogenesis and lipid allocation in adipose tissue [56]. CAMP plays a vital role in the interaction between muscle contraction and calcium signaling and contributes to glycogenolysis to meet the energy requirements for muscles [57]. Previous studies have shown that cAMP pathway activation in muscles contributed to improved myofiber size and muscle strength [58]. The TCA cycle is a significant metabolic network that provides energy and supports precursors for synthetic processes and reducing factors that fuel energy production [59]. The TCA cycle is associated with IMF deposition in castrated cattle [60]. P53 was vital in regulating metabolic pathways, and its increased expression was found in obese rats and humans [61].

In addition, the qRT-PCR results showed that the expression of *TNNI1* and *MYL9* differed significantly between male and female in low-IMF pigs, which further indicated that *TNNI1* and *MYL9* might be reliable biomarkers. Especially in the *TNNI1*, the expression was significantly up-regulated in the male pigs in qRT-PCR and RNA-seq results.

### 4.2. High-IMF Group Analysis

In the high-IMF group, BPs of DEGs were mainly enriched in inflammation/ immunization-related signaling pathways, such as the regulation of interleukin-8 (IL-8) production and leukocyte migration. IL-8 facilitates macrophage infiltration in adipose tissue, which induces topical and systemic inflammation [62]. Obese individuals with a worsened glycometabolic profile and increased inflammation linked to fat tissue show higher expression of IL-8, an inflammation-related adipokine [63,64]. KEGG enrichment analysis revealed that DEGs were mainly involved in lipid metabolism-related processes, such as the MAPK signaling pathway, cytokine–cytokine receptor interaction, and JAK-STAT signaling pathway; cellular immune regulation-associated pathways, such as the T cell receptor signaling; and cell function regulation-associated pathways, such as the PI3K-Akt signaling. The MAPK pathway is one of the critical signaling systems that mediates cellular responses to the external environment and has an influential role not only in the proliferation and differentiation of adipocytes, but also in the development of muscle fibers, affecting sarcomeric traits by regulating the muscle fiber type [65,66]. A recent study also showed that MAPK signaling might be crucial for angiogenesis and adipogenesis in the male pig’s subcutaneous adipose tissue [67]. Cytokine–cytokine receptor interaction signaling pathways influence IMF deposition by regulating the upstream *PPAR* signaling pathway in lipid metabolism [68,69]. The JAK-STAT signaling pathway plays a vital role in IMF deposition by regulating myocyte differentiation and proliferation, thereby affecting meat quality in pigs and goats [70,71].

In addition, ten hub genes were selected from the PPI network. The functional analysis results of the above DEGs were supported by the hub genes’ considerable associations with immunity and inflammation regulation-related processes, such as the regulation of IL-8 production, T cell receptor signaling pathway, and primary immunodeficiency. Subsequently, *CD4*, *CD2,* and *AOC2* genes were selected as potential sex-specific biomarkers in high-IMF pigs and further investigated. Among them, *AOC2* may be the most promising sex-specific biomarker in high-IMF pigs. *CD4*, which encodes the *CD4* membrane glycoprotein of T lymphocytes, is primarily expressed in T lymphocytes. Some investigations have shown the association between CD4^+^ T cells and the progression of obesity and obesity-related diseases, suggesting the role of *CD4* in controlling immune and adipose tissue [72]. Activated or decreased CD4^+^ T cells in adipose tissue play a vital role in maintaining the pro-inflammatory state in obesity [73]. Previous studies have shown that CD4^+^ T cells might be activated through the expression of the main histocompatibility complex class II and co-stimulatory membrane receptors of adipocytes to satisfy the adaptive immune response that adjusted adipose tissue inflammation [74]. Additionally, the function of CD4^+^ T cells is regulated by adipocyte-derived factors, such as adipokines and lipids [75]. *CD2*, a glycoprotein expressed on the surface of most human T cells and natural killer cells, is crucial for mediating cell attachment in both T-lymphocytes and in signal transmission [76]. *CD2* is crucial during the initial steps of the immune response. Several studies have documented the role of anti-CD2 monoclonal antibodies in interfering with afferent immunity, inducing CD2-antigenicity and downregulating *CD3*, *CD4*, and *CD8* cell surface expression [77]. *AOC2* is involved in oxidation by cytochrome P450 and meta-pathway biotransformation phases I and II, which are associated with median neuropathy and amyotrophic lateral sclerosis. The up-regulated *AOC2* expression during adipocyte differentiation in vitro showed that this gene might play a vital role in adipogenesis [78,79]. Studies on gene transcriptional changes of human monoamine oxidases and adipocyte-specific markers during adipogenesis in human bone marrow mesenchymal stem cells observed that the expression pattern of *AOC2* was similar to that of the adiponectin and *PPARc* biomarkers [80]. However, there are few studies on muscle formation and IMF deposition-related *CD4*, *CD2*, and *AOC2* in male and female high-IMF pigs at present. The GSEA functions of *CD4*, *CD2*, and *AOC2* included IMF deposition and immunity-related pathways, such as the JAK-STAT and T cell receptor signaling pathways; this was consistent with the results of the above DEG functional analysis.

In addition, both RNA-seq and qRT-PCR results revealed that *AOC2* was significantly up-regulated in male pigs of the high-IMF group, which further indicated that *AOC2* might be a reliable sex-specific biomarker.

Although sex-specific biomarkers have been identified, and some genes have been validated in previous studies, some limitations of the current study must be noted. First, the sex-specific biomarkers were obtained through a bioinformatics method; biomarker expression levels should be validated via the assessment of larger sample sizes and more pig breeds via more accurate methods. Second, the specific functions of biomarkers need to be verified using overexpression or knockdown methods in a cell or animal.

## 5. Conclusions

In summary, we screened potential sex-specific biomarkers in male and female pigs in the low-IMF (*TNNI1*, *MYL9*, and *SERPINC1*) and high-IMF groups (*CD4*, *CD2*, and *AOC2*). In the low-IMF group, *TNNI1* was evaluated as the most valuable biomarker. *AOC2* was evaluated as the most valuable biomarker in the high-IMF group. The different expression patterns of DEGs, hub genes, and potential sex-specific biomarkers were shown in the male and female pigs with different IMF level, with more genes enriched in low-IMF pigs for muscle fiber and organ-formation-associated processes, such as cardiac chamber morphogenesis, ventricular cardiac muscle tissue development, and muscle contraction, but with more genes enriched in high-IMF pigs for metabolic processes, inflammation, and the immune system, such as the MAPK signaling pathway, cytokine-cytokine receptor interaction, JAK-STAT signaling pathway, the regulation of IL-8 production, and the T cell receptor signaling pathway. The results of the present study showed that IMF deposition was closely related to sex. In addition, there were differences in the molecular and regulatory mechanisms of IMF deposition between male and female pigs with varying IMF levels. These findings provided new insights into the molecular mechanisms of pig IMF deposition and meat quality improvement.

## Figures and Tables

**Figure 1 genes-14-01695-f001:**
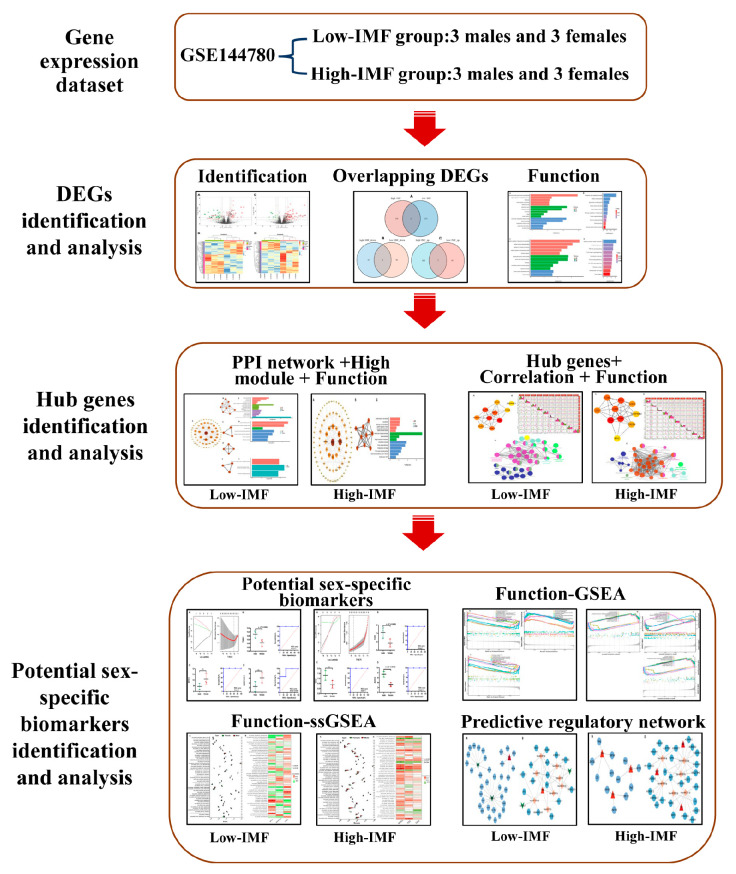
Flow chart of the bioinformatics analysis in the present study.

**Figure 2 genes-14-01695-f002:**
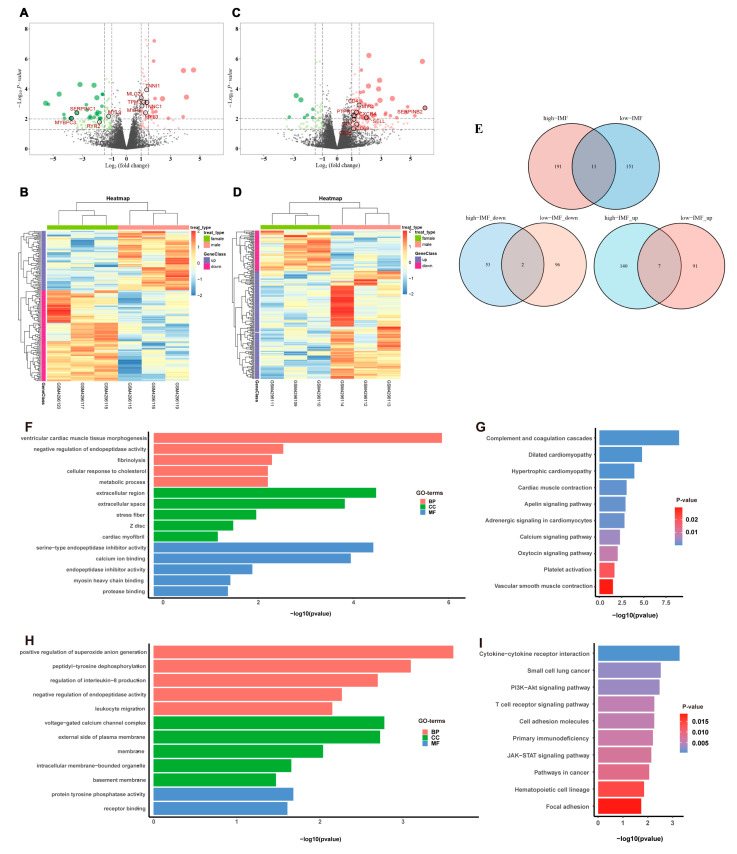
Analysis of DEGs. DEGs were defined with |log2FC| > 1.5 and *p*-value < 0.05. (**A**) Volcano plot of the expression level of DEGs between male and female pigs in low-IMF group. Red dots represent a high expression of genes and green dots represent a low expression of genes. The dots with black circles were hub genes. (**B**) Heatmap of the expression level of DEGs between male and female pigs in low-IMF group. The abscissa indicates the sample names, and the ordinate shows the gene names. High expression of genes is shown in violet and low expression of genes is shown in red. (**C**) Volcano plot of the expression level of DEGs between males and females in high-IMF group. (**D**) Heatmap of the expression level of DEGs between males and females in high-IMF group. (**E**) The Venn diagram of DEGs. (**F**) The results shown by GO of DEGs in low-IMF group. (**G**) The results shown by KEGG of DEGs in low-IMF group. (**H**) The results shown by GO of DEGs in high-IMF group. (**I**) The results shown by KEGG of DEGs in high-IMF group.

**Figure 3 genes-14-01695-f003:**
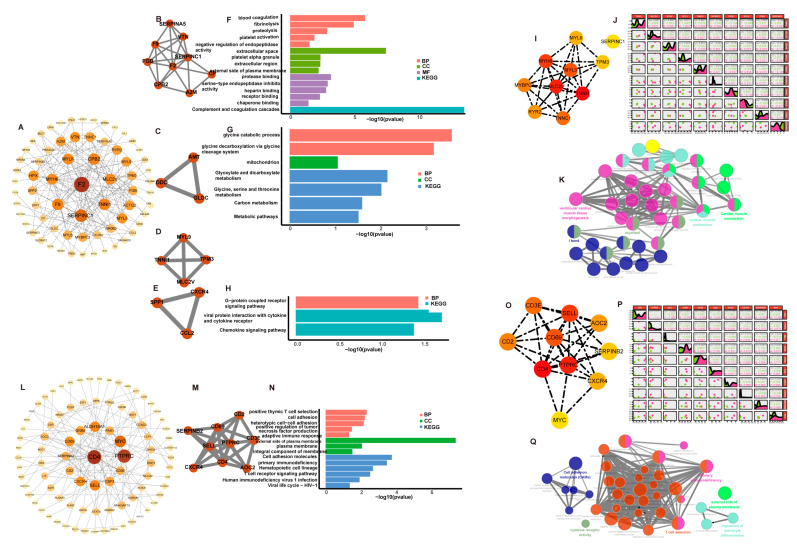
Hub genes identification and analysis. (**A**) A PPI network including 152 nodes and 165 edges in low-IMF group. (**B**–**E**) Highly correlated module with the high score in low-IMF group. (**F**–**H**) Enrichment analysis results of these four clusters; the genes of third clusters were not enriched for any GO category or KEGG pathway in low-IMF group. (**I**) Ten hub genes were selected in the PPI network in low-IMF group. (**J**) The correlation of the ten hub genes in low-IMF group. (**K**) Significantly functional enrichment pathway of ten hub genes in low-IMF group. (**L**) A PPI network including 184 nodes and 150 edges in high-IMF group. (**M**) Highly correlated module with the high score in high-IMF group. (**N**) Enrichment analysis results of cluster in high-IMF group. (**O**) Ten hub genes were selected in the PPI network in high-IMF group. (**P**) The correlation of the ten hub genes in high-IMF group. (**Q**) Significantly functional enrichment pathway of ten hub genes in high-IMF group.

**Figure 4 genes-14-01695-f004:**
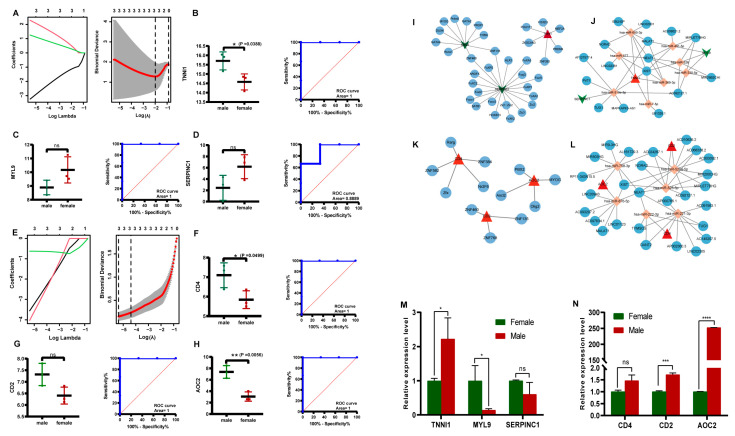
Potential sex-specific biomarkers’ screening and validation. (**A**) LASSO regression model in low-IMF group. (**B**–**D**) Expression of potential sex-specific biomarkers in male and female pigs in low-IMF group; ROC curves of potential sex-specific biomarkers ((**B**) TNNI1; (**C**) MYL9; (**D**) SERPINC1). (**E**) LASSO regression model in high-IMF group. (**F**–**H**) Expression of potential sex-specific biomarkers in male and female pigs in high-IMF group; ROC curves of potential sex-specific biomarkers ((**F**) CD4; (**G**) CD2; (**H**) AOC2). (**I**) TFs-potential sex-specific biomarker network in low-IMF group, where the red triangles symbolize up-regulated potential biomarkers, the green arrows symbolize down-regulated potential biomarkers, and the blue nodes denote TFs. (**J**) lncRNA-miRNA-mRNA (potential biomarker) network in low-IMF group, where the red triangles symbolize up-regulated potential biomarkers, the green arrows symbolize down-regulated potential biomarkers, the blue nodes denote lncRNAs, and the orange nodes denote miRNAs. (**K**) TF-potential sex-specific biomarker network in high-IMF group. (**L**) lncRNA-miRNA-mRNA (potential biomarker) network in high-IMF group. Validation of mRNA expression levels of potential sex-specific biomarkers by qRT-PCR in semimembranosus muscles of Saba pigs for (**M**) low-IMF group, (**N**) high-IMF group. The symbol * means significant difference, **, *** and **** mean extremely significant difference, and ns means no significant difference.

**Figure 5 genes-14-01695-f005:**
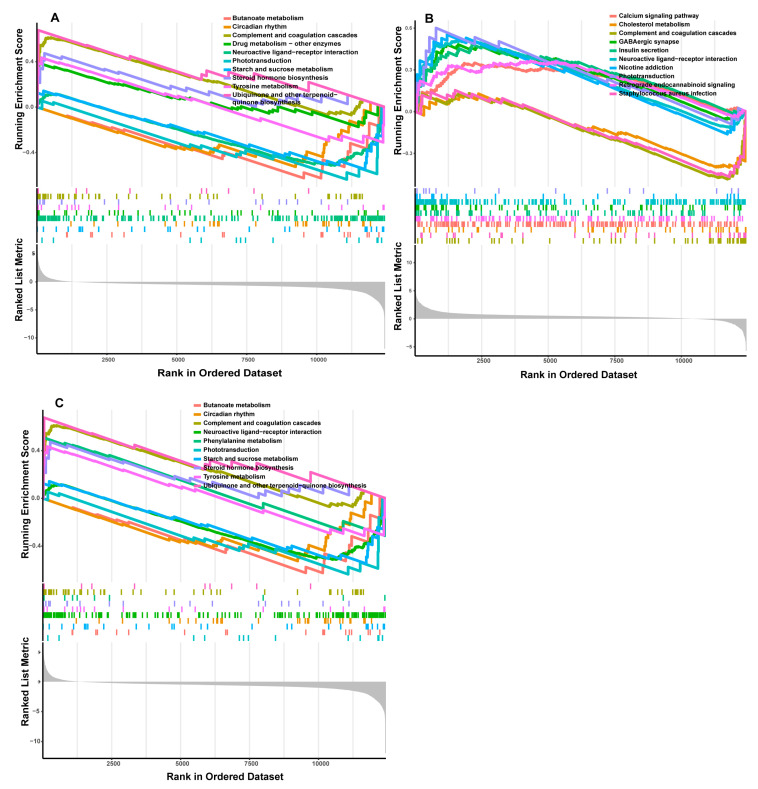
GSEA functional analysis of potential sex-specific biomarkers in low-IMF group. ((**A**) TNNI1; (**B**) MYL9; (**C**) SERPINC1).

**Figure 6 genes-14-01695-f006:**
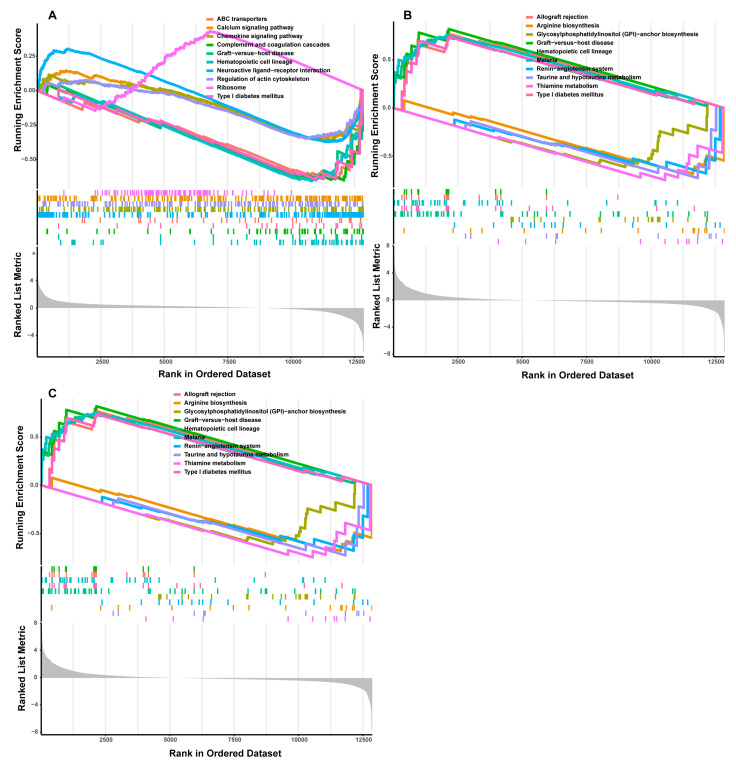
GSEA functional analysis of potential sex-specific biomarkers in high-IMF group. ((**A**) CD4; (**B**) CD2; (**C**) AOC2).

**Figure 7 genes-14-01695-f007:**
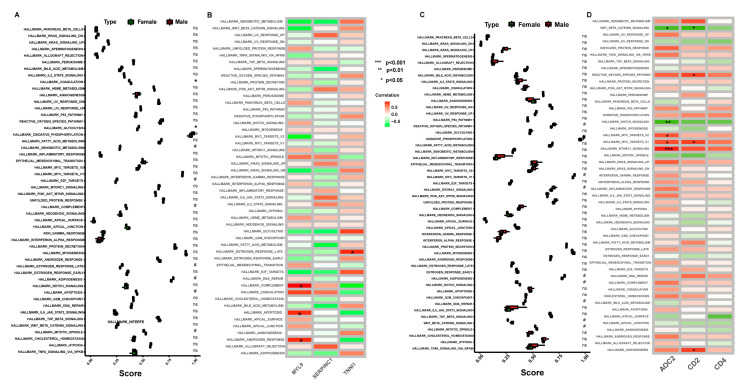
Association analysis of potential sex-specific biomarkers with the hallmark gene sets. (**A**) ssGSEA score of hallmark gene sets and expression characteristics between male and female pigs in low-IMF group. (**B**) Correlation between hallmark gene sets and potential sex-specific biomarkers in low-IMF group. (**C**) ssGSEA score of hallmark gene sets and expression characteristics between male and female pigs in high-IMF group. (**D**) Correlation between hallmark gene sets and potential sex-specific biomarkers in high-IMF group.

**Table 1 genes-14-01695-t001:** Sample information of the GEO dataset.

Group	Total	Male	Female
Low-IMF	6	3	3
High-IMF	6	3	3

**Table 2 genes-14-01695-t002:** Identified overlapping DEGs.

Group Overlapped	Common Genes
High- vs. low-IMF DEGs	*MYL2*, *CXCR4*, *GADD45GIP1*, *DCLRE1B*, *CALCR*, *ITIH1*, *GCKR*, *LYZ*, *GSTO2*, *PLPPR3*, *AKR1D1*
High- vs. low-IMF down-regulated DEGs	*GCKR*, *GSTO2*
High- vs. low-IMF up-regulated DEGs	*MYL2*, *CXCR4*, *DCLRE1B*, *CALCR*, *LYZ*, *PLPPR3*, *AKR1D1*

**Table 3 genes-14-01695-t003:** IMF content of Saba pigs. The results are expressed as mean ± SD. Different IMF level within the same gender were compared.

Group	IMF Content (%)		Weight (kg)	
	Low-Group	High-Group	Low-Group	High-Group
Male	3.63 ± 0.62B	11.27 ± 1.58A	99.67 ± 6.55	108.67 ± 7.41
Female	4.30 ± 1.21b	12.10 ± 2.69a	99.50 ± 4.95	105.83 ± 5.90

The IMF content was compared between the low and high-IMF groups, and the capital and lowercase letters indicate *p* < 0.01 and *p* < 0.05, respectively.

## Data Availability

The datasets (GSE144780) for this study can be found in the in the public NCBI GEO database (https://www.ncbi.nlm.nih.gov/geo/query/acc.cgi?acc=GSE144780).

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
