# Peer review of "Identification of Potential Sex-Specific Biomarkers in Pigs with Low and High Intramuscular Fat Content Using Integrated Bioinformatics and Machine Learning"

_genes, 2023, doi:10.3390/genes14091695_

Round 1

Reviewer 1 Report

In this study, the authors identified sex-specific biomarkers that are related to intramuscular fat content using bioinformatic and molecular analysis. The present study gives us the new information especially in the field of animal production and meat quality. There are some parts of the description that need attention before publication. Some concerns must be revised and clarified in order, for the manuscript, to be published. 

Main concerns:

Lines 57-58: Add references showing the pork quality is different between males and females. IMF content in males is higher than females?

Lines 64-65: “The research flowchart of the present analysis is shown in Figure 1.” Move to Materials and Methods.

Line 157: Please explain the criteria of high or low IMF pigs, and please provide the information in detail.

Lines 284-291: Immune and inflammatory-related pathways are not sex-specific. Are they higher in males or females? How are they related to IMF?

Line 363: The gene expression dataset is originated form Italian Large White pigs. In this study, the authors validate the results using Saba pigs. The IMF content varies in different breeds. Are there any other genes/biomarkers involved in IMF deposition in different breeds? 

Minor concerns:

Please use high resolution figures.

Moderate editing of English language required

Reviewer 2 Report

General Comments

The gene expression data set was retrieved and produced using semimembranosus muscle (SM) samples from Italian Large White pigs (six males and six females). Overall, this is a very well-written paper. The results were presented clearly and concisely. These were adequately discussed but could still be improved upon. However, I have some major concerns that the authors need to address.

The authors need to justify using just twelve pigs in this study. Is this an adequate number to draw any logical conclusion from this study?

The authors need to provide a detailed description of the population of origin and of the 12 samples used in this study, even if mentioned elsewhere. The environmental conditions including details of their feeding regime have not been disclosed, yet very important.

The authors have written the word upregulated in this manuscript with or without a hyphen. Both are acceptable, but it is best to be consistent and adopt one. The same applies to downregulated.

Specific comments

Abstract

Lines 11-13 There’s a need to make a few changes here, ‘was’ is in the wrong tense and ‘evidences’ is an uncountable noun and shouldn’t be in plural.

Introduction

Line 33 The spelling of flavor is a non-British variant. For consistency, please change it to British English spelling. Other words like fiber (line 219), color (line 424) and a few others have been used in the manuscript. This should be checked and implemented accordingly.

Line 58 The preposition ‘of’ is incorrectly used here; change it to ‘in’.

Lines 64-67 The research flowchart of the analysis is presented in the wrong place. It should be presented under materials and methods, which could be explained.

Materials and Methods

Line 78 The original study has been approved by the Institutional Review Board; any reference number given for this?

Line 132 Biomarker does not agree in number with other words in this phase. The same applies to line 137. This should be changed to make the biomarker agree in number with the words used in the sentence(s).

Line 157 The study's methodological details must be understood independently of previous studies or external sources; please consider briefly describing how the IMF content of SM was measured by the Soxhlet extraction method.

Line 159 As noted on line 157 above the study's methodological details must be presented and understood than just referring to the instructions from the manufacturers. Please consider briefly describing how Total RNA was extracted from SM samples using the RNA sample total Extraction Kit.

Line 160 It is equally essential to describe the reverse transcription following the manufacturer’s instructions for your readers to follow. The same applies to the qPCR assay in line 162.

Line 165 The word ‘though’ does not fit this context. This word should be replaced with a different one. Suggestion: Through.

Lines 189, 193 The adjective ‘differential’ is modifying ‘expressed’ instead of a noun or pronoun. Use an adverb to modify a verb or adjective. Suggestion: differential.

Line 219 The spelling of fiber is a non-British variant. For consistency, please change it to British English spelling.

References

This is well-written, and the MDPI guidelines for referencing were followed except that the authors’ initials must be separated from the first name of the next author. 

These have been highlighted above.

Reviewer 3 Report

The work aimed to identify potential sex-specific biomarkers in male and female pigs with different IMF levels. Bioinformatics analysis and machine learning algorithms were used to identify differentially expressed genes (DEGs) that promote IMF deposition. This scientific work is about a very important theme, and discussed problems deserve a lot of scientific attention. The title and abstract are clear. This study could be of interest to the Journal's readers. The manuscript develops the subject logically and effectively, and scientific content of the paper justify its length. The section “Materials and Methods” requires thorough revision, as in the present form it is unacceptable. Chapter 3.9. “IMF Content of SM in Saba pigs” I suggest moved to the Materials and methods. Results and Discussion are well-written. Proposed conclusions are correct and documented by presented results. The tables and figures are necessary. References are actuality and in the topic of the present research.

Minor corrections should be taken into account before publication:

L 74-75  “6 low and 6 high-IMF group” correct  “6 low and 6 high-IMF animals”

Table 1 you wrote that the experiment was conducted on 24 pigs, whereas in the line 145 that there were 30 pigs. What is the correct number?

Why were only 24 pigs included in the study instead of 30, this would have led to an improvement of the statistical results?. The effort for analysis would be comparatively small.

Why did you choose only 6 low and 6 high-IMF animals ?

L 102 “max. Depth=100” correct “max. depth=100”.

Round 2

Reviewer 2 Report

See the attachment please
